# A Pineal Germinoma with Rapid Enlargement following Tumor Resection

**DOI:** 10.3390/diagnostics13233579

**Published:** 2023-12-01

**Authors:** Chia-Jung Hsu, Hsiang-Chih Liao, Dueng-Yuan Hueng, Kuan-Yin Tseng

**Affiliations:** 1Department of Neurosurgery, Tri-Service General Hospital, National Defense Medical Center, Taipei 114, Taiwan; roya1b2c3@gmail.com (C.-J.H.); tomliao0506@gmail.com (H.-C.L.); hondy2195@yahoo.com.tw (D.-Y.H.); 2Division of Neurosurgery, Department of Surgery, Tri-Service General Hospital Songshan Branch, National Defense Medical Center, Taipei 105, Taiwan

**Keywords:** pineal germ cell tumors, radiotherapy, chemotherapy, β-human chorionic gonadotropin

## Abstract

The natural course of pineal germ cell tumors (GCTs), particularly their post-operative progression, is not well understood. We report a rare case of pineal region GCT showing rapid enlargement within 2 weeks following surgical resection. A young adult male presented with progressive headache and diplopia for several weeks. Although elevation of β-human chorionic gonadotropin (β-HCG) and α-fetoprotein (AFP) levels suggested that a large pineal mass lesion observed on magnetic resonance imaging (MRI) might be a β-HCG/AFP-producing tumor, whether the mass was truly a GCT remained unclear. We performed an endoscopy-assisted suboccipital infratentorial approach with removal of the tumor that was diagnosed as germinoma via histopathological investigation. During the week preceding chemotherapy, the patient’s consciousness rapidly worsened. MRI showed that the residual pineal germinoma had enlarged and even compressed the tectum and thalamus. Emergency chemotherapy and radiotherapy were prescribed, and the patient received invasive ventilation for respiratory failure. Unexpectedly, the patient recovered within a short period. Importantly, total regression of the pineal germinoma, accompanied by β-HCG and AFP levels returning to normal range, was observed 4 months after chemotherapy. These phenomena suggest that the rapid enlargement of the pineal germinoma, which might be induced by aggressive surgical cytoreduction, responds well to chemoradiotherapy.

A 24-year-old male, working as a boilerman on a warship, with a hairy body surface, presented to our emergency department with progressively worsening headaches and diplopia for 2 weeks. Initial computed tomography (CT) of the brain revealed a mass with contrast enhancement in the posterior area of the third ventricle. Magnetic resonance imaging (MRI) of the brain revealed a heterogeneously enhancing pineal region mass lesion with obstructive hydrocephalus as shown in Figure 1. Serum samples showed extremely high levels of tumor markers, such as β-HCG and AFP. Surgical intervention with external ventricular drainage was performed as a strategy for CSF diversion, followed by an endoscopy-assisted suboccipital infratentorial approach with the removal of the tumor for cytoreduction, as shown on the brain CT 3 days post-operatively (Figure 2). The frozen section suggested a pineal germinoma, in accordance with its histomorphological pattern. Immunohistochemical staining revealed high expression of glial fibrillary acidic protein, SALL-4 (sal-like protein 4), and placental-like alkaline phosphatase (PLAP) in the pineal tumor, which was confirmed to be a pineal germinoma (Figure 3).

In the first postoperative week, the patient was capable of following instructions, even with the endotracheal tube and EVD in place, as a precaution against postoperative cerebral edema. During this period, the Karnofsky performance score (KPS) was 60. However, 2 weeks after surgical intervention, the Glasgow coma scale deteriorated to 7 from the normal level. Moreover, the KPS dropped to 20, accompanied by Parinoud’s syndrome. Thus, an emergency endotracheal tube was inserted due to impaired consciousness. A repeated MRI of the brain showed an enlarging heterogeneous enhancing mass, measuring 4.4 × 3.0 × 4.0-cm, occupying the pineal region and tectal plate and displacing the vein of Galen and inferior sagittal sinus with perifocal edema in bilateral thalamus and right basal ganglion as shown in Figure 4, compared with preoperative imaging. After explaining the outcome and risk of the subsequent therapeutic strategy to his family, chemotherapy with etoposide (600 mg) and cisplatin (140 mg) followed by ifosfamide (8000 mg) was administered in divided doses for 4 days with simultaneous intensive ventilation support. Unexpectedly, the patient recovered rapidly after chemotherapy. The tumor size dramatically decreased, as observed on MRI scan, with three cycles of chemotherapy. Moreover, radiotherapy with a total dose of 24 Gy in 22 fractions was administered over 3 weeks to the gross tumor before chemotherapy, followed by a local boost total dose of 16 Gy in 10 fractions over 2 weeks to the residual tumor after chemotherapy.

MRI performed 3 months after chemotherapy showed complete remission of the pineal germinoma as shown in Figure 5. The patient’s neurological function recovered well. Twelve months after chemotherapy, at outpatient follow-up, MRI demonstrated no tumor relapse, accompanied by β-HCG and AFP levels within the normal range (Figure 6). The patient was completely independent and actively participated in military career.

Primary germ cell tumors represent 0.4% of CNS tumors. Germinomas, the most prevalent subtype, often arise in the pineal region. They can also appear in suprasellar, bifocal, basal ganglia, or multifocal areas [1,2,3]. Surgical biopsy via open or stereotactic biopsy for tissue diagnosis plays a significant role in the diagnosis and management of pineal tumors because of the lack of specific imaging characteristics. Intracranial germinomas respond well to radiation, with combined chemotherapy and radiotherapy yielding excellent outcomes [4,5,6,7]. If a pineal region tumor is homogeneously enhanced and has the classic appearance of germinoma on MRI with isointensity relative to CSF and elevated β-HCG and AFP, which is essential for the diagnosis of germinoma, a test dose of 5 Gy may be administered. If the tumor shrinks, the diagnosis of germinoma is almost certain and radiotherapy may be continued without surgery [8,9]. However, this approach may obey institutional policies, and some centers do not offer chemotherapy and radiotherapy to patients without histological confirmation. Patients with pineal germinoma often present with obstructive hydrocephalus. Surgical interventions like ventricular peritoneal shunt or endoscopic third ventriculostomy alleviate symptoms and offer a chance for a pathological diagnosis [10] However, the endoscopy-assisted biopsy diagnosis of a pineal tumor, given its histological heterogeneity, seems to have limitations in accurately representing the entire tumor. Therefore, in this case, the decision to proceed with tumor removal was made not only to achieve cytoreduction but also to obtain the most extensive specimen possible for a comprehensive histological diagnosis. Even with the best efforts secure sufficient tumor tissue for diagnostic confirmation, there remains a possibility that the tumor could be a mixed cell-type germ cell tumor.

The reason for pineal germinoma’s swift growth post-surgery is unclear. In our observation, the tumor had a significant blood supply, potentially leading to rapid growth. Devascularization is recommended during tumor removal [11] Pineal germinomas risk CSF seeding, with most relapses in the periventricular area. Whole-brain radiation results in 8% recurrence, while focal radiation has 23%. CSF flow influences both tumor seeding and growth [12,13]. We report a case of a pineal germinoma manifesting with dramatic size changes. Given germinomas’ responsiveness to chemotherapy and radiotherapy, early aggressive chemotherapy is advised, especially when large tumors and neurological issues are not due to obstructive hydrocephalus. Despite the tumor’s size and vascularity, chemotherapy and radiotherapy post-diagnosis are prioritized over surgical reduction, even in comatose patients.

## Figures and Tables

**Figure 1 diagnostics-13-03579-f001:**
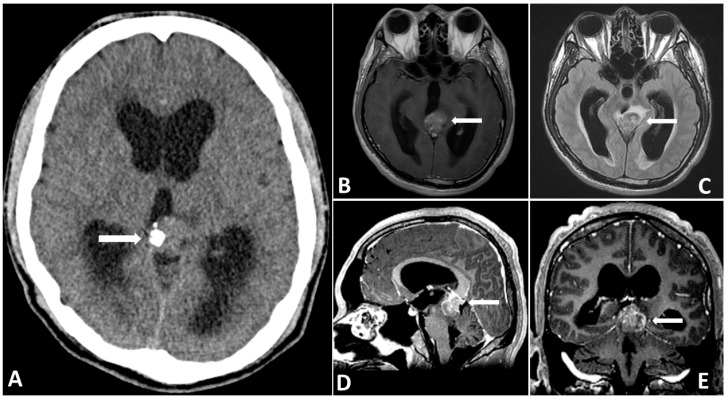
Preoperative CT (**A**) showing a partially calcified mass in the pineal region causing hydrocephalus. Preoperative MRI showing that the (**B**) tumor mass is hyperintense on T2-FLAIR images. (**C**–**E**) Gadolinium-enhanced MRI, revealing a large pineal region tumor with heterogenous contrast enhancement.

**Figure 2 diagnostics-13-03579-f002:**
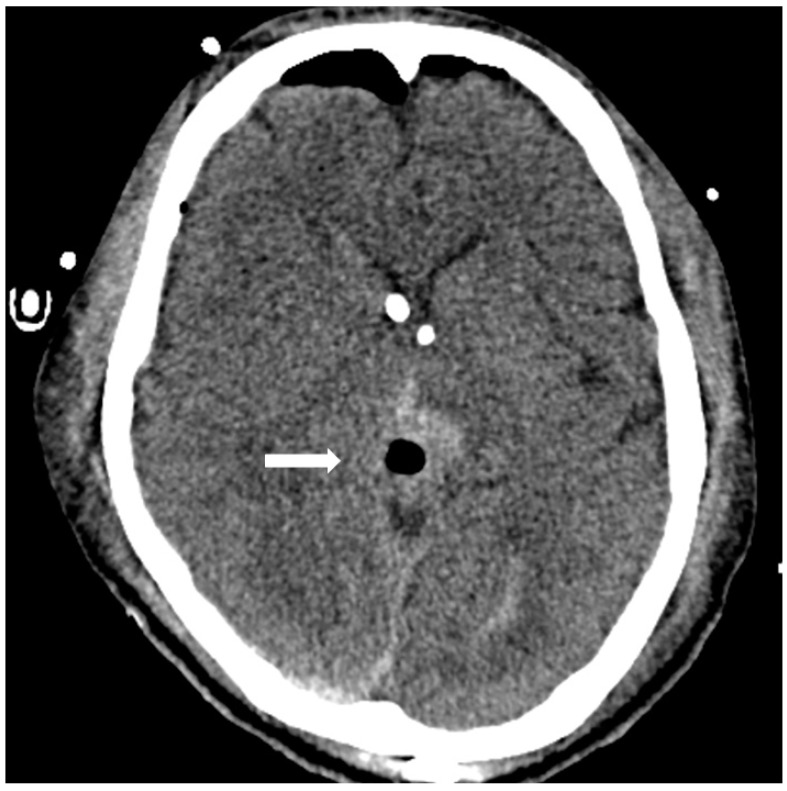
Postoperative CT showing removal of pineal tumor via an endoscopy-assisted occipital transtentorial corridor and retention of a bilateral external ventricular drainage tube.

**Figure 3 diagnostics-13-03579-f003:**
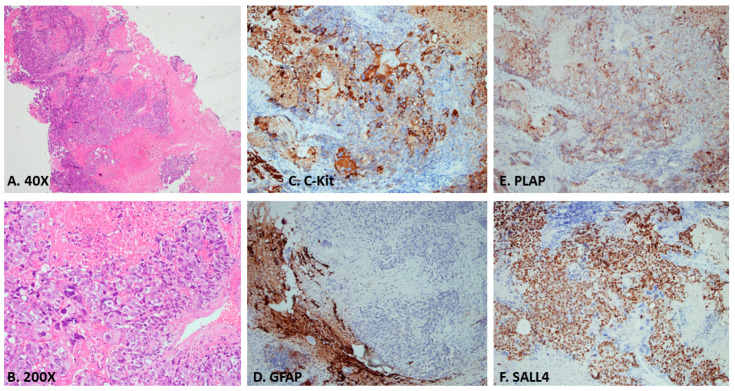
(**A**,**B**) Photomicrograph of a section from the tumor showing germinoma cells with pale cytoplasm, well defined cell membranes, and round central nuclei. Immunohistochemical staining of germinomas showing (**C**) intense C-Kit expression. (**D**) Negative GFAP expression in this pineal germinoma case. (**E**) Intense placental alkaline phosphatase (PLAP) expression. (**F**) The nucleus is intense react to sal-like protein 4 (SALL4).

**Figure 4 diagnostics-13-03579-f004:**
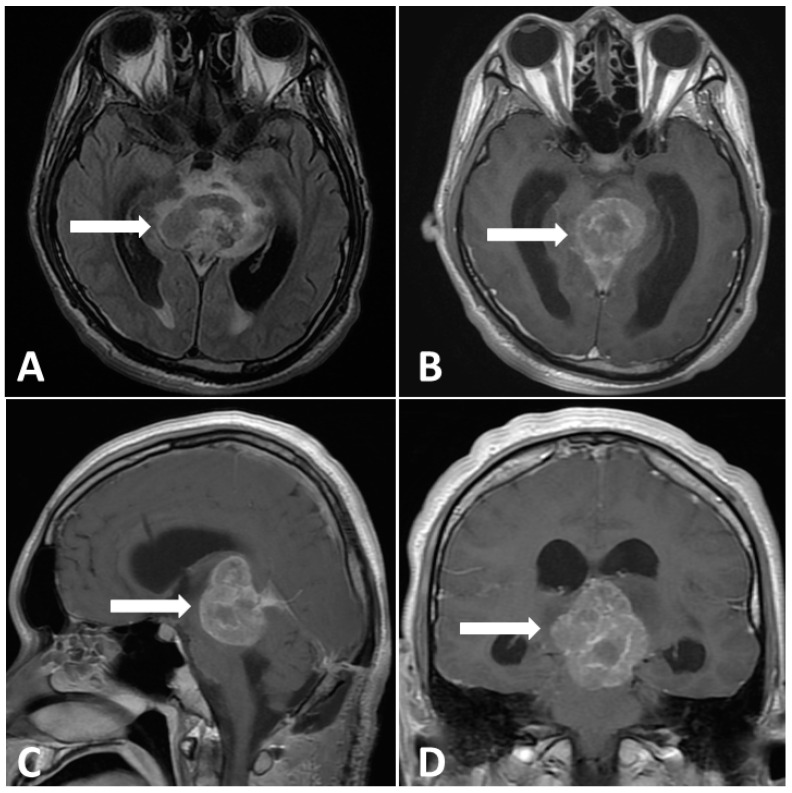
Postoperative MRI performed 2 weeks after surgery. (**A**) Enlargement pineal tumor with perifocal edema on T2-FLAIR images. (**B**–**D**) Gadolinium-enhanced MRI, revealing a large pineal region tumor with heterogenous contrast enhancement.

**Figure 5 diagnostics-13-03579-f005:**
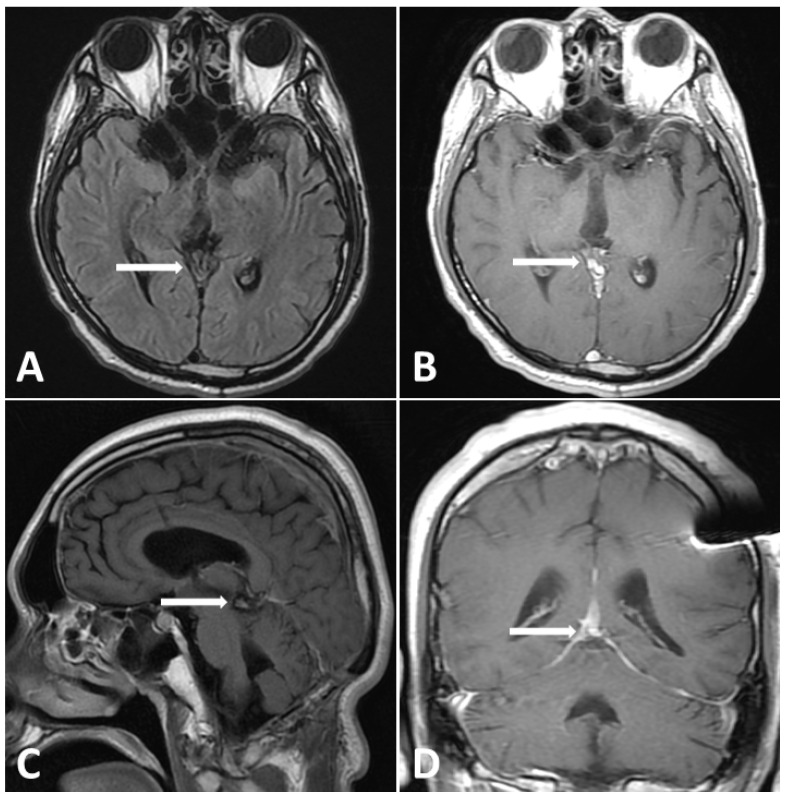
Follow-up MRI (3 months post treatment). (**A**) T2-FLAIR images showing evidence shrinkage of lesion without perifocal edema. (**B**–**D**) Gadolinium-enhanced MRI, revealing dramatic shrinkage of previous pineal tumor.

**Figure 6 diagnostics-13-03579-f006:**
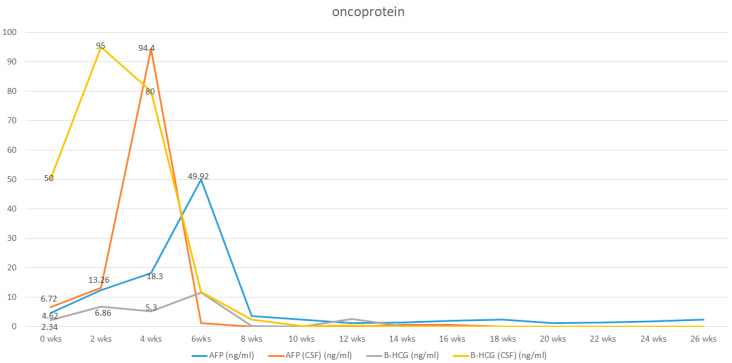
Elevation of tumor markers (Serum and cerebrospinal fluid levels of hCG and AFP) at initial diagnosis, which was elevated after surgical intervention and dramatically dropped after chemotherapy.

## Data Availability

The data presented in this study are available on request from the corresponding author. The data are no publicly available due to restrictions of privacy.

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
