# Peer review of "A Pineal Germinoma with Rapid Enlargement following Tumor Resection"

_diagnostics, 2023, doi:10.3390/diagnostics13233579_

Round 1

Reviewer 1 Report

Comments and Suggestions for Authors

The description of the case is not complete as values of Beta-HCG and AFP  at diagnosis (before surgery), after surgery and after chemotherapy are not reported clearly; only a graph is reported  without arrows and dates reporting the time of evaluation of markers. Post operative imaging  is represented only by a slice of CT scan and than not sufficient to estimate the entity of tumour removal. An MRI is necessary.  The decision to attempt tumour removal  for cytoreduction in a case with elevated tumours markers (both Beta-HCG and AFP)  is, according to international guidelines not appropriate versus hydrochephalus treatment +/-minimal biosy followed by  "immediate" chemotherapy  as elevated tumours markers indicate a Malignant Secreting Mixed Germ Cell Tumours  and not Pure Germinoma

The histological diagnosis of pure Germinoma  if the tumour sample is small how can happen after endoscopic biopsy,  cannot be representative of the entire biology of the tumour. 

In this case the presence of heterogeneity of CE, calcification etc. visible at preoperative imaging is suggestive not of pure germinoma but od secreting mixed germ celll tumour. 

I

Author Response

Response to Reviewer 1 Comments

Thank you very much for taking the time to review this manuscript. My coauthors and I were gratified to learn your suggestion to our manuscript. Hopefully our paper may be acceptable for publication in “Diagnostics” after revisions.

In this letter, we have detailed a point by point response to reviewers and will also cite where changes and additions were made in the revised manuscript.

Point-by-point response to Comments and Suggestions for Authors

Comments 1: The description of the case is not complete as values of Beta-HCG and AFP at diagnosis only a graph is reported without arrows and dates reporting the time of evaluation of markers.

Response 1: Thank you for pointing this out. We agree with this comment. Accordingly, we included the test values and dates for Beta-HCG and AFP to enhance the accuracy of the depiction of the tumor marker changes in the image."  (uploaded Ward file)

Comments 2: Post operative imaging  is represented only by a slice of CT scan and than not sufficient to estimate the entity of tumor removal. An MRI is necessary.

Response 2: Indeed, we totally agree the reviewer’s suggestion. However, on 1 day after surgery, the patient was still on the E-T tube and dependent on ventilation support in the ICU unit. In our hospital, Radiology Dr. requested the patient to wean off the endotracheal (E-T) tube and subsequently undergo a post-operative MRI scan. In this case, an emergent CT image was taken due to the suspected occlusion of the extraventricular drainage (EVD) tube. Two weeks later, since the patient’s level of consciousness deteriorated, a contrast-enhanced brain MRI was promptly performed to assess the situation. Therefore, in this CT scan images, we just want to show that we did tumor cytoreduction without any rebleeding or other OP-related complications.

Comments 3: tumor removal for cytoreduction versus hydrocephalus treatment +/-minimal biopsy followed by immediate chemotherapy for Mixed Germ Cell Tumors or Germinoma.

Response 3: The patient presented with Parinaud syndrome, characterized by upgaze palsy and Collier's sign, suggestive of a compression at the level of the midbrain tectum. Therefore, we conducted a tumor removal procedure with the goal of cytoreduction and obtaining tissue for a histological diagnosis to the greatest extent possible. While awaiting the pathology report, the tumor grew several times in size over a two-week period, leading to a deterioration in the patient's level of consciousness. Fortunately, chemotherapy had just commenced at this time. Through aggressive chemotherapy and radiotherapy, we were able to suppress the growth of the tumor.

Comments 4: The histological diagnosis cannot be representative of the entire biology of the tumour.

Response 4: We agree with this comment. Nevertheless, returning to a previous recommendation, a minimal biopsy followed by immediate chemotherapy, which the review suggested, appeared to be less certain in representing the tumor. Therefore, the decision to proceed with tumor removal was taken not only achieving cytoreduction but also obtaining the most extensive specimen possible for a comprehensive histological diagnosis. While our best efforts secure sufficient tumor tissue for this diagnostic confirmation, there remains a possibility that the tumor could be a mixed cell-type germ cell tumor.

Sincerely

Reviewer 2 Report

Comments and Suggestions for Authors

A Pineal Geminoma with Rapid Enlargement Following Tumor Resection

The presented case report demonstrates a patient suffering from pineal germinoma, which was removed surgically though suboccipital infratentorial approach after implantation of an external ventricular drainage from both sides (see Figure 2) in to treat a hydrocephalus. The serum sample detected extremely high level of tumor markers.

A massive enlargement of the germinoma after 2 weeks was noticed. Chemotherapy and radiotherapy were performed immediately, which decreased the germinoma effectively. 

First, the course of this demonstrated disease is not uncommon. In cases of germinoma with preoperative high tumor markers (Beta-HCG, AFP). The most common treatment for germinomas in pinealis reason is a combination of radiotherapy and chemotherapy. The hydrocephalus has treated with a third ventriculostomy endoscopically. 

Nevertheless, this manuscript shored a dramatic course of germinoma in the pinalis region. Therefore, this case report has some right to publish in diagnostics. 

Author Response

Response to Reviewer 2 Comments

Thank you very much for dedicating time to review our manuscript. Your expert feedback serves as a driving force for our academic work and clinical practice in treating patients. Furthermore, we believe that this paper will capture the interest of your journal's readership, as it sheds light on the natural course and post-operative progression of pineal region germ cell tumors, which is currently not well-documented. This case provides valuable insights that could enhance our understanding of pineal region germ cell tumors (GCTs). The content of this case report aligns with the scope of the journal Diagnostics. My coauthors and I are encouraged by the news that our paper may be considered acceptable for publication in Diagnostics, subject to revisions.

Sincerely

Reviewer 3 Report

Comments and Suggestions for Authors

The authors present an interesting case of a pineal germinoma with enlargement of tumor resection which is well presented with illustrative images. It gives valuable information about the possible course of this disease. Every neurosurgeon should be aware that immediate radiation/chemotherapy should be considered if the serum levels of beta-HCG/AFP are highly suspicious for germinoma as in the present case.  

However, there are some unclear issues:

There was only a postoperative CT-scan and it seems to show residual tumor.  Did you achieve a gross total resection during surgery? Why did you not perform an MRI as the gold standard to confirm the extent of resection? You referred to the vascular problems that might occur in this region, which could have been detected with MRI also.  

The second question is why did you wait 2 weeks with deterioration of the patient until you performed the first MRI?  Or was there an earlier MRI which is not displayed? 

Could you please comment on that?

Comments on the Quality of English Language

The language is sufficient and only requires minor corrections.

Author Response

Response to Reviewer 3 Comments

Thank you very much for taking the time to review this manuscript. My coauthors and I were gratified  to know that there is a possibility for our paper to be published in Diagnostics, subject to the necessary revisions. In this letter, we have provided a detailed response to each of the reviewers' comments and have indicated where we have made changes and additions in the revised version of the manuscript.

Point-by-point response to Comments and Suggestions for Authors

Comments 1: postoperative CT-scan and it seems to show residual tumor.  Did you achieve a gross total resection during surgery? Why did you not perform an MRI as the gold standard to confirm the extent of resection? You referred to the vascular problems that might occur in this region, which could have been detected with MRI also.

Response 1: Thank you for pointing this out. Due to the limited field of view during the endoscopy-assisted suboccipital infratentorial approach and the adhesion of the tumor's texture to surrounding tissues, we were unable to achieve total removal of tumor. We could only accomplish cytoreduction and obtain histopathological confirmation of the diagnosis to the best of our ability.

Comments 2: why wait 2 weeks with deterioration of the patient until you performed the first MRI?  Or was there an earlier MRI which is not displayed? 

Response 2: Postoperatively, the patient was placed in the intensive care unit for postoperative care, and a CT scan was obtained due to a suspected blockage in the external ventricular drainage tube. During the period awaiting the pathology report, the tumor grew several-fold within two weeks, leading to a worsening of the patient's level of consciousness. Consequently, a contrast-enhanced Brain MRI was performed promptly to evaluate the situation. Fortunately, chemotherapy had already commenced at this point, and with the aggressive administration of chemotherapy and radiotherapy, we were able to achieve suppression of the tumor growth.

Sincerely

Round 2

Reviewer 1 Report

Comments and Suggestions for Authors

I do not have any more suggestions to modify the paper as all my previous comments have been accepted.

Reviewer 3 Report

Comments and Suggestions for Authors

The authors commented each issue and added the requested topics to the manuscript. Therefore, I can recommend the paper for publication in the present form.